# Stainless Steel Surface Nitriding in Open Atmosphere Cold Plasma: Improved Mechanical, Corrosion and Wear Resistance Properties

**DOI:** 10.3390/ma14174836

**Published:** 2021-08-26

**Authors:** Alice O. Mateescu, Gheorghe Mateescu, Adriana Balan, Catalin Ceaus, Ioan Stamatin, Daniel Cristea, Cornel Samoila, Doru Ursutiu

**Affiliations:** 1Horia Hulubei National Institute of Physics and Nuclear Engineering, 30 Reactorului Street, 077125 Magurele, Romania; mateescu@nipne.ro; 23 Nano-SAE Research Centre, Faculty of Physics, University of Bucharest, 405 Atomistilor Street, 077125 Magurele, Romania; catalin@3nanosae.org (C.C.); istarom@3nanosae.org (I.S.); 3Department of Material Science, Faculty of Materials Science and Engineering, Transilvania University of Brasov, 1 Colina Universitatii, 500036 Brasov, Romania; daniel.cristea@unitbv.ro; 4Technical Science Academy of Romania, 125 Calea Victoriei, 010071 Bucharest, Romania; 5Electronics and Computer Department, Faculty of Electrical Engineering and Computer Science, Transilvania University of Brasov, 29 Eroilor Blvd., 500036 Brasov, Romania; udoru@unitbv.ro; 6Romanian Academy of Scientists, 3 Ilfov Street, 050045 Bucharest, Romania

**Keywords:** surface nitriding, open atmosphere, cold plasma, corrosion resistance, hardness, wear

## Abstract

This work presents preliminary results regarding improving the mechanical, wear and protective properties (hardness, coefficient of friction, corrosion resistance) of AISI 304 stainless steel surfaces by open atmosphere cold plasma surface treatment method. Comparative evaluations of the morphological, corrosion resistance, mechanical and tribological properties for different periods of treatment (using N_2_ gas for cold plasma generation in an open atmosphere) were performed. AFM surface analyses have shown significant surface morphology modifications (average roughness, FWHM, surface skewness and kurtosis coefficient) of the treated samples. An improved corrosion resistance of the N_2_ treated surfaces in open atmosphere cold plasma could be observed using electrochemical corrosion tests. The mechanical tests have shown that the surface hardness (obtained by instrumented indentation) is higher for the 304 stainless steel samples than it is for the un-treated surface, and it decreases gradually for higher penetration depths. The kinetic coefficient of friction (obtained by ball-on-disk wear tests) is significantly lower for the treated samples and increases gradually to the value of the un-treated surface. The low friction regime length is dependent on the surface treatment period, with a longer cold plasma nitriding process leading to a significantly better wear behavior.

## 1. Introduction

Stainless steels are widely used in industrial applications due to their durability and high corrosion resistance. However, there are some weaknesses of steel (such as low hardness and poor wear resistance) that need to be addressed. A promising surface engineering technology is nitriding, as it enables the atomic nitrogen diffusion into the superficial surface layers of a material creating a surface with increased properties, e.g., increased hardness, wear and corrosion resistance. These processes are more commonly applied on low-carbon steels, but they are also used on medium-carbon or high-carbon steels that contain metals with ∆G/mol < 0 at T < 100 °C, according to the Ellingham diagram, as well as on titanium, zirconium, aluminum, molybdenum, etc. and their alloys [1].

Regarding the plasma nitriding of Fe alloys, the reactivity of the nitriding media is not due to the high temperature, as is the case for gas nitriding or salt bath nitriding processes, where the temperature grows up to 600 °C, but to the nitrogen gas ionized state (N_2_^+^), that interacts with the surface material. It causes the diffusion of the nitrogen ions inside of the material structure and the occurrence of chemical reaction with components of the Fe alloys [2,3,4,5,6,7]. It is well known that the reaction of nitrogen with the usual components of stainless steel (Fe, Ni, Cr, Mo, Ti, Mn, C, Si, etc.) spontaneously occurs at low temperatures, only if the standard Gibbs energies of the nitride formation in the low temperature range are negative values (∆G < 0) [8]. As one could see in the Ellingham diagrams for selected nitrides, the Gibbs free energy that predicts the direction of a chemical reaction concerning certain chemical compounds of N_2_ (ZrN, TiN, AlN, Nb_2_N, Mg_3_N, Si_2_N_4_, VN, CrN, Cr_2_N and Mo_2_N) has negative values on a large range of temperatures (300–1200 K); these are the same temperatures that are used in the classical ion nitriding as well as in the atmospheric cold plasma ion nitriding [9].

The diffusion process and chemical reaction of nitrogen in a steel material (plain carbon steel, low-alloy steel, high-alloy steel) are accelerated using an increased temperature in all three types of nitriding processes (not only in ion nitriding) by producing different phases, resulting in different phases such as α, γ, γ′, ε phases. The α + ε phases appear at low temperatures, specific for cold plasma treatment in open atmosphere. The α phase is a ferritic steel phase with a BCC (body centered cubic) crystalline structure, while the ε phase is an intermetallic compound with an HCP (hexagonal close packed) crystalline structure [10]. The major components of the Fe-N binary phase diagram that could be formed in an iron nitriding process (FeN, Fe_2_N, Fe_3_N, Fe_4_N, Fe_3_N_4_, Fe_16_N_2_, etc.) exhibit different thermal stabilities as well as varying magnetic properties [11]. To expand, FeN and Fe_2_N are N-rich phases, while Fe_3_N, Fe_4_N, Fe_3_N_4_, Fe_16_N_2_ are N-poor phases_;_ Fe_4_N and Fe_3_N are ferromagnetic phases; Fe_3_N change into the paramagnetic Fe_2_N phase at room temperature when the content of nitrogen increases from 25% to 33%; Fe_4_N phase, that contains around 20% N, has distinct magnetic properties and FCC crystal structure and could be produced by controlled annealing of FeN phase; FeN phase has a low thermal stability because of the weak Fe-N bonding, etc. [11].

Open atmosphere cold plasma surface treatment (OACP-ST) is a relatively new process up-graded to the industrial stage for surface treatment, especially for polymers or textile material treatment [12,13,14]. OACP-ST is making its way towards implementation on metallic materials as well. It was reported that radiofrequency cold plasma treatment, performed in vacuum for 8 h on non-heated C38 carbon steel, resulted in the formation of a uniform structure of the sample surface, with an enrichment in nitrogen and a gradient of hardness versus depth [15].

AISI-304 steel, used in our work, is one of the most known and used materials in the manufacturing of the mechanical components in the automotive, chemical, textile, medical, airspace, etc. industries, due to its favorable mechanical, chemical, thermal and magnetic properties. It has the following chemical composition in wt%: max. 0.08%—C; max. 2%—Mn; max 0.045%—P; max. 0.03%—S; max. 0.75%—Si; max. 0.10%—N_2_; 18–20%—Cr; 8–12%—Ni and Fe in balance [16]. However, the low hardness and deficient wear resistance of AISI 304 stainless steel (304SS) in certain environments limits its application in many fields. Several works report the improvement of their mechanical, tribological and protective properties by vacuum plasma ion nitriding.

R. Valencia et al. showed that the AISI 304 SS was successfully nitrided in vacuum (10^−1^ … 10^−3^ mbar), by using plasma. The Vickers hardness has been increased several times compared to the untreated samples and the higher values were obtained at a higher pressure of the experimental pressure range, while the tribological properties improved substantially below 500 °C without it losing its corrosion resistance [17]. A short-term treatment and low temperature novel nitriding process that uses an “active screen” was reported to be more effective than the classic cathode glow-discharge process, leading to an increase in the surface hardness, in the diffusion depth of the nitride layers and in the abrasive wear resistance [18]. T. Balusamy and his collaborators sustain the capability of the Surface Mechanical Attrition Treatment (SMAT) to increase the depth of the nitrided layer formed on 304 SS [19]. Plasma nitriding was also used on welded joints of 304 SS parts, with good results in terms of surface hardness, phase formation and case depth [20]. AISI 316L subjected to the nitriding process at temperature lower than 450 °C presents a high nitrogen content expanded austenite phase that shows higher hardness and higher pitting corrosion resistance compared to the untreated material, the technological parameters of the nitriding process playing an important role in the hardness and N concentration [21]. Investigating the carbon behavior in low temperature plasma nitriding, M. Tsujikawa et al. indicated an accumulated carbon layer beyond the nitrogen plateau, which means the low temperature plasma nitriding of austenitic stainless steels pushes carbon ahead of the nitride layer [22]. It has been found that the ion nitriding surface treatment improved the fatigue strength and increased the fatigue limit, depending on the case depth [23]. Furthermore, the thickness of the compound layer increased with an increase in the treatment time and temperature for the ion nitrided AISI 4340 steel [24].

To the extent of our knowledge, there are no scientific works reporting on the capability of the OACP-ST to improve the corrosion resistance, hardness or friction coefficient of metals or alloys in general (or of the 304 SS in particular) using only commercial nitrogen without any supplementary gas (e.g., nitrogen and hydrogen gas mixture that is used in classical or vacuum ion nitriding processes). However, some aspects of the OACP-ST influence on 304 SS were investigated [25,26], where it was reported that plasma treatment in open atmosphere has a significant positive effect on the wettability and contact angle as well as on the surface free energy and surface roughness of the treated samples. The hydrophilicity of the treated surface was significantly improved by decreasing the surface contact angle and the treated surfaces exhibited lower carbon content than the untreated one [25,26].

In this paper, we report the nitriding of 304 SS sample surface by open atmospheric cold plasma surface treatment, according to the Patent Application A00261/02.05.2017 of the authors A. O. Mateescu and G. Mateescu, registered at the Romanian State Office for Inventions and Trademarks (OSIM), aiming at the improvement of the corrosion resistance, mechanical and tribological behavior of the treated surfaces. AFM investigations for surface topography, corrosion resistance evaluation and mechanical and tribological results of treated and un-treated samples are presented below.

## 2. Materials and Methods

Three hundred and four SS sheets and commercial nitrogen (5.0) as a working gas (both process and cooling gas) at a pressure of 6 bar and a flow of 1.5 m^3^/h for cold plasma surface nitriding treatment were used. Plasma beam equipment with an open atmosphere cold plasma torch for surface treatment (Plasma Beam Equipment, Dienner Electronic GmbH, Ebhausen, Germany) was employed. The main components of the cold plasma treatment in open atmosphere and the geometrical parameters of the plasma torch for the nitriding process are presented in Figure 1. The diameter of the plasma beam ranged from 10 to 12 mm, while the height of the plasma beam could be selected from 5 to 10 mm. For the x–y movement of the plasma beam (PC controlled values), a CNC equipment type MINI-CNC engraving machine with a variable speed from 0.5 up to 20 mm/s was used. The samples with dimensions of 20 mm × 20 mm × 2 mm were fixed on the CNC Machine and moved in x direction with a speed selected in the range of 0.5 mm/s up to 20 mm/s, at around 6 mm distance from the surface (on z axis), with repetition of movement on a new line in x direction, at 4 mm distance on the y axis. The general duration of the process was:-a total time of the cold plasma treatment of 0.25 h, with moving speed of plasma beam of 9 mm/s for 3 min/line-cycle and 5 line-cycles/sample, for sample 1;-a total time of the cold plasma treatment of 3 h, with moving speed of plasma beam of 108 mm/s for 36 min/line-cycle and 5 line-cycles/sample, for sample 2.

An ultrasonic bath cleaning process of 0.5 h preceded the surface nitriding treatment of the AISI 304 SS samples.

AFM topography was performed with SPM-NTegra Prima AFM (NT-MDT, Russia) in semicontact mode, using a NSG 01 cantilever (resonance frequency: 83–230 kHz, elastic constant: 1.45–15.1 N/m, scan rate 1 Hz, scan resolution of 512 samples per line). The AFM images were recorded on 10 × 10 µm^2^ scan surface. Grain size distributions, surface skewness and coefficient of kurtosis were calculated using NT-MDT Image Analysis 2 software.

Electrochemical measurements were conducted in a conventional three electrode cell: the working electrode (the sample), the corrosion electrode (Pt) and the reference electrode (saturated calomel electrode-SCE) in 0.1 M aqueous NaCl solution. All solutions were prepared from chemically pure (c.p.)/practical grade (p.a.) chemicals (Merck) and bi-distilled water. The corrosion tests are performed in the range ± 100 mV vs. open circuit potential at a scan rate 1 mV/s. The corrosion resistance (polarization resistance) and the corrosion potential are measured from Tafel plot. Measurements were performed using Voltalab PGstat model 301 (Radiometer Analytical, Lyon, France).

The instrumented indentation hardness and instrumented indentation elastic modulus of the untreated and treated 304 SS samples in OACP-ST were determined following the model of Oliver and Pharr [27]. Nanoindentation measurements (CSM Instruments (NHT-2)—Berkovich diamond tip) were performed at different penetration depths in order to qualitatively assess the drop in hardness, caused by the limited nitrogen diffusion depth. As per the ISO 14577 standard (Metallic materials—Instrumented indentation test for hardness and materials parameters), at least 10 indentations were performed on each sample, with 30 s loading/unloading speeds up to the desired maximum indentation load and a 10 s pause between the loading and unloading stages, in order to minimize the creep effect. The friction coefficient of the samples was evaluated by ball-on-disk wear tests, using a Standard Tribometer, from CSM Instruments/Anton Paar (Peseux, Switzerland), in rotation mode. Six mm WC (tungsten carbide) balls were used as friction couples, with a normal applied load of 2 N, a linear speed of 22 cm/s, 3.5 mm and 4.5 mm radius tracks and a 600-m stop condition. The samples were ultrasonically cleaned in isopropanol prior to the wear tests, in order to remove any impurities that might affect the wear test results. The variation of the friction coefficient as function of the test length was the main focus.

## 3. Results

### 3.1. AFM Analysis

Figure 2 shows the AFM micrographs of the surface of bare stainless steel (Sample 0), sample 1 and sample 2 and Figure 3 presents the height histograms of AFM images. Images of all samples were processed and analyzed by means of the offline NT-MDT Image Analysis 2 software. As shown in A–C the films reveal a granular homogenous surface morphology. The following roughness parameters were determined for all samples: average roughness (the arithmetic average of absolute values of the surface height deviations measured from the mean plane), full width at half maximum-FWHM (determined from the Gauss fitting of the histograms), surface skewness and coefficient of kurtosis (see Table 1). The average roughness increases for treated samples of stainless steel up to 31.03 nm. Grain boundaries become sharper and the matrix shows a less smooth appearance. These results could be explained by assuming that nitriding of stainless steel surface leads to the formation of more compact and higher aggregates.

The analysis of the surface morphology in terms of average roughness, skewness and coefficient of kurtosis can be correlated with the experimental measurements of the friction coefficient. The skewness of the profile can be used to distinguish between profiles with similar roughness parameters, but different shape: a symmetrical height distribution has zero skewness and deep scratches give negative values, while high peaks result in positive values. In our case, sample 1 shows a negative surface skewness, indicating deep scratches and sample 2 a positive coefficient, but higher than the bare stainless steel. These results support the idea of grain boundary becoming sharper, with higher aggregates. On the other hand, the kurtosis coefficient describes the sharpness of the profile. The kurtosis coefficient of stainless steel is 3.50 (i.e., a leptokurtoic distribution), corresponding to many high peaks and low valleys. After nitriding the surface, the kurtosis coefficient decreases to 1.32 (sample 1) and 1.43 (sample 2), respectively, which describes a platykurtoic distribution corresponding to the few high peaks and low valleys.

### 3.2. Corrosion Resistance Results

The corrosion potential and the polarization resistance, determined from Tafel plot (see Figure 4), are summarized in Table 2, for bare stainless steel and samples 1 and 2. The stainless steels are usually quite cathodically relative to other alloys, as they exhibit electrode potentials, in saline water, from +500 to −280 mV vs. SCE [26]. The electrode potential (E_corr_) of bare stainless steel was found at −117 mV vs. SCE, with a polarization resistance (R_p_) of 94.84 kΩ/cm^2^. Sample 1 has a corrosion potential more negative than stainless steel, −211 mV vs. SCE, but with a much higher R_p_ of 177.47 kΩ/cm^2^. On the other hand, sample 2 has the lowest corrosion potential, −267 mV vs. SCE, but with a lower R_p_ of 76.37 kΩ/cm^2^. Results of the Tafel extrapolation method show that the treated stainless steel has the lowest current density, which suggests that the nitriding process increases the corrosion resistance of bare stainless steel. The higher polarization resistance of sample 1 indicates that the nitrided stainless steel are more effective in protecting against corrosion. Moreover, the corrosion rate (calculated using Faraday’s law, according to ASTM G-102) is significantly decreasing after the nitriding process, i.e., 0.30 µm/year for sample 1 and 1.06 µm/year for sample 2 compared to the 2.52 µm/year for the bare stainless steel.

### 3.3. Mechanical Parameters: Hardness, Elastic Modulus and Wear Behaviour

Table 3 contains the results obtained from the instrumented indentation analysis. The instrumented indentation hardness (H_it_) and elastic modulus (E_it_) were obtained for multiple penetration depths (h_m_) as a result of 1.5, 2.5 and 5 mN loads (F_m_). One can notice that the highest hardness values are measured at low penetration depths as a direct result of the nitriding process. The variation of hardness as function of penetration depth can be observed in Figure 5. A steady decrease can be observed, towards hardness values associated to the untreated sample. A longer nitriding period results in slightly higher hardness values; however, the diffusion depth is relatively the same, regardless of nitriding duration.

The H/E ratio can provide information about the wear resistance of the material in question [28], while the H^3^/E^2^ ratio gives information about the resistance to plastic deformation.

Inferred from the H/E ratio, seen in Table 3, the nitride samples should exhibit relatively similar values for the dynamic friction coefficient, at least at the beginning of the test. This observation is confirmed by the graph shown in Figure 6. Even though the stop condition for the wear tests was set at 600 m, the graph from Figure 6 shows only the region up to 20 m, to better visualize the differences in the earlier stages of the friction tests. Past this value (20 m), the friction coefficient was stable throughout the entire tests. A low friction regime is noticed for the nitrided samples; however, the distance of this regime is significantly lower for the 0.25 h treated sample. Both the slightly lower hardness and the smaller diffusion depth are responsible for this behavior, due to the fact that the nitrided material is removed easier from the surface of the sample. Once the nitrided material is removed from the 3 h-treated sample, the dynamic friction coefficient values are relatively close to the ones exhibited by the untreated sample. There is a clear correlation between the nitriding period and the wear resistance of the surface, for longer processing stages, one should expect a better wear behavior.

## 4. Conclusions

Surface nitriding treatment by cold plasma in open atmosphere at low temperature was performed on AISI 304 SS samples.

Surface morphology analyzed by AFM shows that the open atmosphere with cold plasma treatment determines an increase of the roughness from around 4 nm (for untreated stainless steel—sample 0) to around 30–31 nm (samples 1 and 2).

The corrosion resistance has improved for the nitrided samples as the corrosion potential has decreased from −117 mV for untreated bare stainless steel (sample 0) down to −211 mV for 0.25 h treatment in cold plasma (sample 1) and −211 mV for 3 h treatment (sample 2).

Diffusion of the nitrogen in the sample depth with nitriding process of 304 SS samples, by open atmosphere cold plasma surface treatment method with commercial nitrogen (5.0), is put in evidence in addition to the indentation test when the highest hardness values were measured at the lowest penetration depth and the lowest hardness value was measured at the highest penetration depth.

The coefficient of friction with very low values (less than 0.20) was obtained for the treated samples, but these low values were observed only for a relatively short sliding length (lower than 4 m), that increased with longer treatment process.

Improvement of surface quality (hardness, coefficient of friction, corrosion resistance) of the AISI 304SS samples by cold plasma treatment is strongly influenced by the treatment duration and technical parameters of the plasma jet (intensity, diameter and length of plasma jet). The temperature increasing of the metallic treated surfaces by cold plasma is also very reduced for long time treatments, without using external heating sources of the metallic samples. The low cost of the equipment for open atmosphere cold plasma treatment compared to that for vacuum ion nitriding is another reason for using this new surface treatment for different practical applications (e.g., the enhanced surface characteristics of large dimension industrial tools.)

## Figures and Tables

**Figure 1 materials-14-04836-f001:**
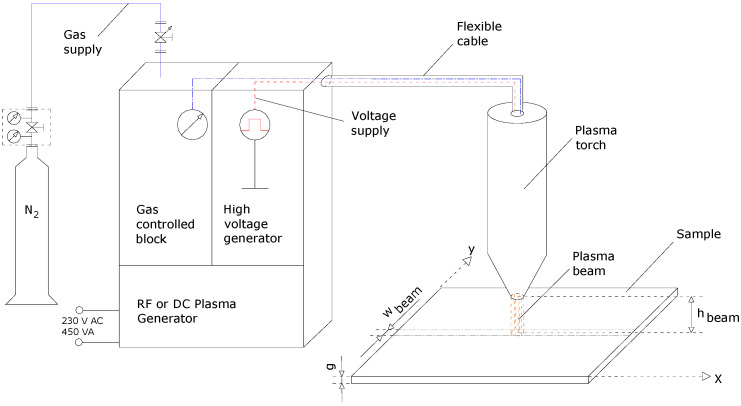
Plasma beam equipment for nitriding process of AISI 304 SS sample and geometrical parameters of the plasma beam.

**Figure 2 materials-14-04836-f002:**
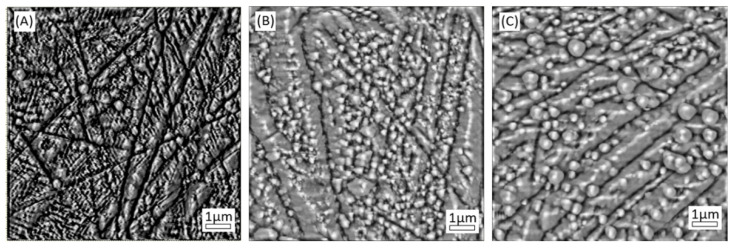
AFM micrograph of: Sample 0 (**A**), sample 1 (**B**) and sample 2 (**C**). Scan size 10 µm × 10 µm.

**Figure 3 materials-14-04836-f003:**
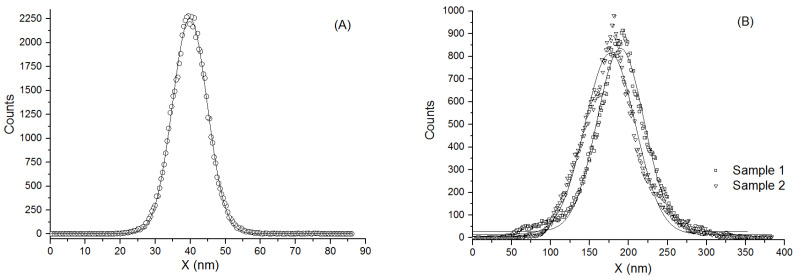
Height histograms of AFM images presented in Figure 2 of: Sample 0 (**A**), sample 1 and sample 2 (**B**).

**Figure 4 materials-14-04836-f004:**
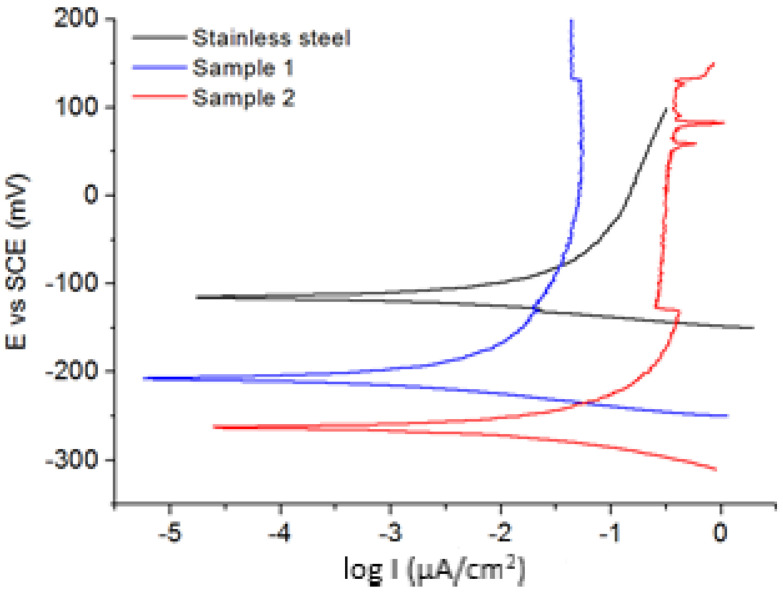
Polarization curve obtained in 0.1 M NaCl for bare stainless steel, sample 1 and sample 2.

**Figure 5 materials-14-04836-f005:**
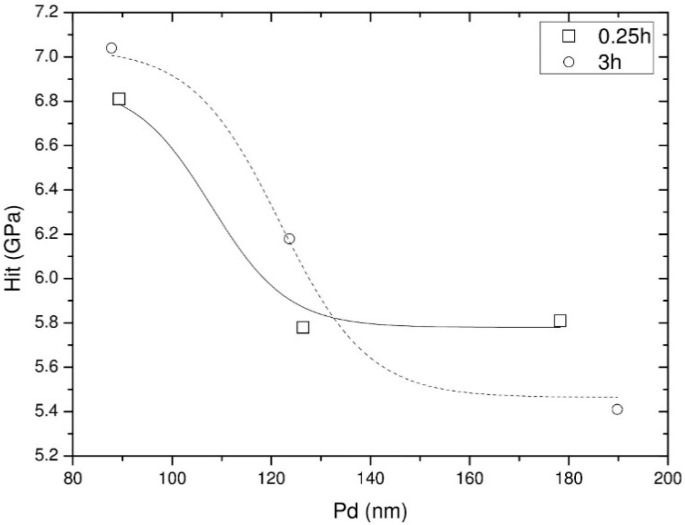
The variation in hardness as function of penetration depth.

**Figure 6 materials-14-04836-f006:**
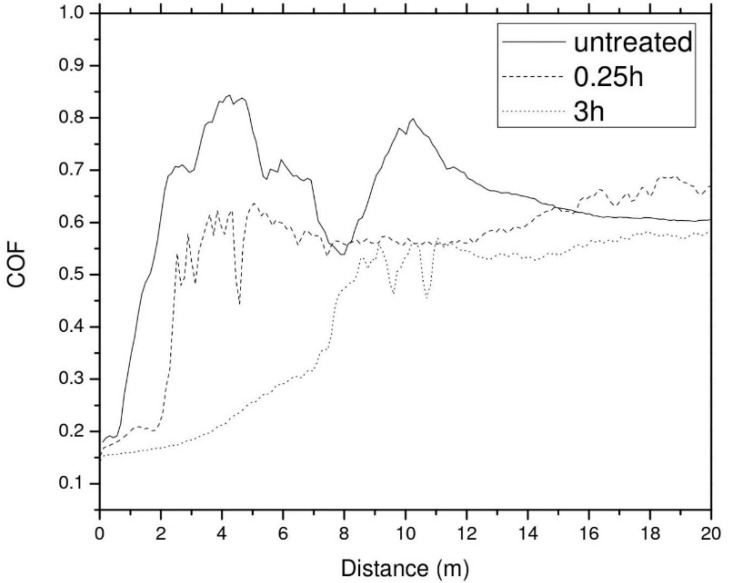
Dynamic friction coefficient variation, as a function of sliding distance.

**Table 1 materials-14-04836-t001:** The roughness parameters of un-nitrided and nitrided AISI 304 SS.

Sample	Average Roughness (nm)	FWHM (nm)	Surface Skewness	Coefficient of Kurtosis
Sample 0 (un-treated bare stainless steel)	4.14	11.61 ± 0.04	0.37	3.50
Sample 1 (0.25 h nitriding treatment)	31.03	71.04 ± 0.57	−0.30	1.32
Sample 2 (3 h nitriding treatment)	30.07	76.38 ± 0.75	0.63	1.43

**Table 2 materials-14-04836-t002:** Corrosion parameters for bare and nitriding stainless steel in 0.1 M NaCl determined by Tafel extrapolation method.

Sample	E_corr_ (mV_SCE_)	j_corr_ (μA/cm^2^)	β_a_ (mvdec^−1^)	−β_c_ (mVdec^−1^)	R_p_ (kΩ/cm^2^)	CR (μm/year)
Sample 0 (untreated bare stainless steel)	−117	0.26	468.1	27.5	94.84	2.52
Sample 1 (0.25 h nitriding treatment)	−211	0.03	40.3	26.0	177.47	0.30
Sample 2 (3 h nitriding treatment)	−267	0.10	59.7	41.2	76.37	1.06

**Table 3 materials-14-04836-t003:** Surface mechanical characteristics as a result of nanoindentation analysis.

Sample-Load No.	H_IT_ (O&P) (GPa)	E_IT_ (O&P) (GPa)	H/E	H^3^/E^2^	h_m_ (O&P) (nm)	F_m_ (mN)
	Mean	Mean			Mean	
1–3	5.81 ± 0.63	199.170 ± 38.65	0.028	0.0050	178.259 ± 8.77	5
1–2	5.78 ± 1.44	204.880 ± 48.47	0.026	0.0039	126.410 ± 12.41	2.5
1–1	6.81 ± 1.08	198.167 ± 18.58	0.034	0.0080	89.315 ± 7.13	1.5
2–3	5.41 ± 0.76	189.047 ± 25.10	0.028	0.0040	189.774 ± 13.74	5
2–2	6.18 ± 1.30	193.362 ± 27.63	0.031	0.0063	123.650 ± 13.11	2.5
2–1	7.04 ± 1.38	204.107 ± 23.28	0.034	0.0083	87.813 ± 8.82	1.5

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
