# Peer review of "Stainless Steel Surface Nitriding in Open Atmosphere Cold Plasma: Improved Mechanical, Corrosion and Wear Resistance Properties"

_materials, 2021, doi:10.3390/ma14174836_

Round 1

Reviewer 1 Report

In the introduction, I have several recommendations for ionic nitriding in general. The statement on line 109 is true, but for ion nitriding to work in this way, it is necessary to select a suitable ratio of gases in the nitriding atmosphere. The gas ratio must prevent the formation of a compound layer, which in turn reduces the fatigue limit. Or it is necessary to mechanically abrade. Related to this is the statement on line 110 about the thickness of the compound layer. The statement is again true depending on the ratio of the main gases. However, it should not lead to the idea that this dependence is uniform or linear. The dependence is definitely nonlinear and the increase in thickness with time slows down significantly. If the author really means compound line on line 110, it is unsuitable with regard to the previous mention of fatigue. But if he thought of a diffusion layer, then its thickness increases the fatigue strength.

In Materials and Methods, the symbolism of the three dots between the numbers is not apparent on lines 135 - 137. Whether it is a range of values, extreme values or something else. I think it should be made clear what this is all about. The description of the ball-on-disk method on line 178 lacks values about test parameters, such as track diameter, rotation speed or measurement time, or distance during the test. This is then related to Fig. 6 in the Results section, where it is not clear whether the COF value is already stable after 20 m. Or, significant value fluctuations continue as the values adjust. If those 20 m can be considered as adjust values? Next on line 180 is a formal error where the dot at the end of the sentence is missing. (… 2 N. The samples…)

There is no agreement in the results in Tab. 3 and a description (line 256) in the designation H / E, H3 / E3.

The conclusion is relatively brief and does not sufficiently describe the nitriding method with regard to practical use in engineering, production, or any technique, for example.

At line 286, there is a formal error in the word plsma (plasma is correct). 

Author Response

First of all, we would like to thank to the referee for the dedicated time to review our draft.

The pertinent suggestions and questions were considered in the new version of the manuscript. We refer to them following each point raised by the reviewer.

Reviewer 2 Report

Z

Author Response

(The authors gave the same response as above.)
